# Genomic Signatures Reveal Breeding Effects of Lulai Pigs

**DOI:** 10.3390/genes13111969

**Published:** 2022-10-28

**Authors:** Rui Cao, Jian Feng, Yuejin Xu, Yifei Fang, Wei Zhao, Zhenyang Zhang, Zhe Zhang, Meng Li, Qishan Wang, Yuchun Pan

**Affiliations:** 1Department of Animal Science, School of Agriculture and Biology, Shanghai Jiao Tong University, Shanghai 200240, China; 2Hainan Institute, Zhejiang University, Building 11, Yongyou Industrial Park, Yazhou Bay Science and Technology City, Yazhou District, Sanya 572025, China; 3College of Animal Sciences, Zhejiang University, Hangzhou 310058, China; 4Jinan Laiwu Pig Industry Technology Research Institute Co., Ltd., Jinan 271100, China; 5Key Laboratory of Livestock and Poultry Resources Evaluation and Utilization, Ministry of Agriculture and Rural Affairs, Hangzhou 310030, China; 6Hainan Yazhou Bay Seed Lab, Yongyou Industrial Park, Yazhou Bay Sci-Tech City, Sanya 572025, China

**Keywords:** introgression, systematic crossbreeding, pig genome, *F_st_*, rIBD

## Abstract

In Chinese pig populations in which crossbreeding is used, these animals show a level of weakness compared with their original purebred ancestors. For instance, in the Lulai pig, a newly developed Chinese breed that is raised on the basis of the Laiwu pig (a Chinese indigenous breed with exceptionally high intramuscular fat content) and the Yorkshire pig using a method of systematic crossbreeding, both their market acceptance and performance are inferior. To reveal the practical role of these admixed breeds and traditional systematic crossbreeding methods at the genomic level, we explored population structure, genetic signatures, and introgression. We conducted this study based on the SNP chip data of 381 Lulai pigs, 182 Laiwu pigs, and 127 Yorkshires, which showed deficient genome coverage during our study. Therefore, we further selected the Genotyping by Genome Reducing and Sequencing (GGRS) method, which has a high density and suitable genome coverage as a supplement. We applied the GGRS data of 38 Lulai pigs, 75 Laiwu pigs, and 75 Yorkshires. In terms of the SNP chip data, by *F_st_* analysis, we detected 782 significantly different genes between Lulai pigs and Yorkshires, including 3 major genes associated with growth (LEPR) and meat quality (SCD and TBC1D1), and we detected 426 significantly different genes between Lulai pigs and Laiwu pigs. With rIBD, we detected 12 genomic regions that included 182 genes that Yorkshires introgressed to Lulai pigs, and we detected 27 genomic regions that included 229 genes with a major gene (SCD) that Laiwu pigs introgressed to Lulai pigs. Regarding the GGRS data, we detected 601 significantly different genes between Lulai pigs and Yorkshires by *F_st_* analysis, including 3 major genes associated with growth and fat deposits (IGF2 and FTO) and with hair color (KIT), and we detected 634 significantly different genes between Lulai pigs and Laiwu pigs, including 3 major genes related to their body composition (MYPN), hair color (KIT), and ear size (PPARD). By rIBD, we detected 94 deep sections that included 363 genes that Yorkshires introgressed to Lulai pigs, and we detected 149 deep sections that included 727 genes with a major gene (ESR1) that Laiwu pigs introgressed to Lulai pigs. Altogether, this study provides both insight into the molecular background of synthesized breeds of Lulai pigs and a reference for the evaluation of systematic crossbreeding in China.

## 1. Introduction

Most Chinese indigenous pig breeds are characterized by their good meat quality and large litter size, as well as slow growth and low lean rates. Western pig breeds, such as the Yorkshire (YY), are characterized by fast growth and high lean rates but have a relatively lower meat quality. Hence, to combine the advantages of both Chinese indigenous and western pig breeds, the traditional crossbreeding method has been widely used in the past century to develop new breeds. This crossbreeding strategy, which is based on ancestor herds and generation-by-generation selection, was originally proposed in Canada. This strategy prevailed in Japan at the end of 1960s and was called “systematic synthesis”, which Mikami translated into “closed-herd breeding over several generations” [1]. After being introduced to China, it was called “polyancestor-based crossbreeding by selection in successive generations” (PCSSG), and in the past 50 years, it has become the leading method in China for developing new breeds. Since then, more than 60 new breeds have been developed by PCSSG. The Lulai pig breed (LU) is a typical newly developed breed that was synthesized using YY pigs and Laiwu pigs (LW). The LW pig is an indigenous Chinese pig breed that is famous for its desirable meat quality (especially for its high intramuscular fat content) [2]. However, with the further development of marketization, some problems regarding LU pigs and other newly synthesized breeds have been found, such as moderate performance, a lack of competitiveness, etc. This has led to researchers evaluating and exploring the real effects of PCSSG.

PCSSG consists of the following stages: first, a group of basic founders is selected; second, the selected group is closed; and finally, further selection and breeding in the closed group are performed according to production performance, body appearance, blood origin, etc. However, the practical operation of PCSSG in the past has involved the use of the selection-based phenotype, so many new breeds developed by this method have not performed well. Therefore, we must evaluate the effectiveness of PCSSG, and we need to determine whether newly developed breeds have reached the breeding goal and inherited the advantages of the breeding materials. We used population structure, selection, and introgression signature analysis in this study to find the results.

Recently, high-throughput sequencing techniques have been widely applied in the genomic research of pigs, which has enables the feasible and reliable detection of genome-wide selection signatures [3]. SNP chip is a commonly used method with high accuracy and efficiency. However, through a follow-up analysis, we found that the genome coverage of SNP chip data was deficient, so we further selected the Genotyping by Genome Reducing and Sequencing (GGRS) method, which has high density and suitable genome coverage as a supplement. This method was also successfully applied to evaluate genetic diversity in some Chinese indigenous pig breeds [4]. In relation to the analyses of identification of selection signatures, the *F_st_* method has been widely used based on genetic differentiation between populations [5,6,7]. Relative identity by descent (rIBD) is a method used to calculate the relative section of two populations’ haplotypes in their offspring, which can accurately reveal introgression signatures [8,9].

Using these analysis methods, we aimed to provide further understanding of PCSSG and insight into the molecular background of the synthesized Lulai pig breed.

## 2. Methods and Methodology

### 2.1. Sample Preparation and Sequencing

We sampled the ear tissues of 381 LU and 182 LW pigs from an original Laiwu conservation farm in Laiwu City, Shandong Province. We collected the ear tissues of 128 Yorkshire pigs (YY) from a commercial pig farm (Fujian Guanghua).

We extracted the genomic DNA from the ear tissues with a commercial kit (Lifefeng Biotech, Co., Ltd., Shanghai, China). In this study, we genotyped the genomic DNA samples of 381 LU pigs and 182 LW pigs using KPS Porcine Breeding SNP-Chip v1 (Compass Biotechnology, Beijing, China), and we genotyped 127 YY pigs using GeneSeek Genomic Profiler Porcine HD BeadChip (Neogen Corporation, Lansing, MI, USA). As a supplement, we genotyped 38 LU, 75 LW, and 75 YY pigs using the GGRS platform.

We used next-generation sequencing quality control (NGSQC) Toolkit v2.3 software to filter the sequencing data mentioned above. Then, we merged the SNP chip data [10], and we marked different locations as missing. We imputed missing genotypes with BEAGLE v4.1 software [11]. Then, we performed quality control procedures for the imputed data with PLINK (population-based linkage) v1.9 software with a minor allele frequency (MAF) filter criterion of lower than 0.05 [12]. In addition, we excluded reads that we could not map.

Regarding the GGRS data, after using NGSQC Toolkit v2.3 to process the quality, we used BWA (Burrows–Wheeler Alignment) software v2.3 to locate the sequencing reads from the porcine reference genome (Sscrofa 11.1) [13]. We combined the segments that we mapped to the genome and obtained SNPs using Genome Analysis Toolkit (GATK) and Samtools software v1.8, separately, to extract the coherent SNPs [14,15]. We used BEAGLE v4.1 and PLINK v1.9 for imputation and quality control with the same SNP chip data standard.

### 2.2. Population Structure

Based on all the data described above, we performed principal component analysis (PCA) using GCTA v1.24, and we plotted the first two PCs via the ‘ggplot2’ R package (Vienna, Austria) [16,17]. In addition, we investigated the population structure using ADMIXTURE v1.30 software, and we computed the genomic components of the best hypothetical ancestor number K using a Bayesian model [18].

### 2.3. Signature of Selection

To detect the selected signatures of LU, LW, and YY pigs, we performed the genetic differentiation method *F_st_*. We defined the significance threshold as the top 1% of all *F_st_* values. An *F_st_* value above 1% represents potential outlier values. We considered the selection signatures detected by *F_st_* to be the candidate loci under selection [19]. We used the GALLO R package to exhibit the results of the QTL enrichment analysis [20].

### 2.4. Functional Annotation of Candidate Genes

We used the Ensembl database (Sscrofa 11.1; http://asia.ensembl.org/index.html; accessed on 15 January 2022.) to identify candidate genes that flanked these candidate loci upstream and downstream in 50 kb regions. To further analyze the functions of these identified genes, we used the Annotation, Visualization and Integrated Discovery (DAVID) database for enrichment analysis [21]. We considered Gene Ontology (GO) terms with *p*-values of <0.01 and Kyoto Encyclopedia of Genes and Genomes (KEGG) pathway terms with *p*-values of <0.05 to be significant results. In addition, with a QTL region length of less than 1 Mb, we used the pig QTL database (SS_11.1) (https://www.animalgenome.org/cgi-bin/QTLdb/SS/index; accessed on 27 January 2022) for the annotation of significant selection signatures.

### 2.5. Detection of Introgression

To investigate which genes LU pigs inherited from their parent populations, LW and YY pigs, we performed rIBD with several procedures. First, we used BEAGLE v4.1 to determine the SNP linkage phase, and then we used the fast IBD function of BEAGLE v3.1 to analyze the IBD fragments between individuals. We repeated this process ten times to improve the reliability of the results. Second, we combined the results of the 10 calculations [22]. Third, by dividing the genome into bins with a length of 10 kb, we counted the number of fragments that shared IBD fragments between each group of the three breeds indicated above (cIBD), and we recorded the total number of shared IBD fragments that could appear in a pair-wise comparison between the two breeds (tIBD). Then, we calculated the number of standardized shared IBD fragments as nIBD = cIBD/tIBD, where tIBD is the product of the number of the two breeds. Fourth, depending on the value of nIBD, we calculated the relative number of shared IBD fragments (rIBD) in certain bins. After separately standardizing the rIBD for different groups, we considered results that deviated by more than 2 standard deviations to be a significant standard. Finally, regarding the significant introgression segments from the parent populations, we used the BiomaRt tool of the Ensembl database within each of the identified introgression segments. For these gene sets, we analyzed enrichment using DAVID software to explore possible functional changes that might have resulted from significant introgression. 

## 3. Results

### 3.1. Sequencing and Detection of SNPs

In total, we generated more than 610 million reads (100 bp). We retained a final set of 79,739 SNPs (minor allele frequency ≥ 0.05) from the SNP chip results of 692 pigs and 375,025 SNPs (minor allele frequency ≥ 0.05) with an average depth of 7 (Phred quality ≥ 20) from the GGRS results of 188 pigs. The nucleotide diversity of the nonoverlapping windows of 105 bp showed a suitable distribution within the whole genome, except for the sex chromosomes in the GGRS results (Figure 1).

### 3.2. Population Structure

Regarding the SNP chip data, the PCA results calculated using the GATK are shown in Figure 2a. According to PC1 and PC2, we could well-distinguish the LW and LU Chinese indigenous pig breeds from the western Yorkshire pig breed. The developed LU pig breed was in a dispersed state between the LW and YY pig breeds, which is consistent with the hybrid identity of LU pigs. Although we could not clearly identify some parts of the LU and LW pig populations, this may be the reason for the excessive mixing of LW pigs in the crossbreeding process.

Regarding the GGRS data, the PCA results calculated using GATK are shown in Figure 2b. We could distinguish the Chinese breed from the western breed, but for the LU and LW pig populations, we could not separate some individuals to a certain degree. In the condition of only two principal components, a part of the LU and LW pig breeds showed a mixed state, which was similar to the SNP chip results, indicating that some excessive introgressions of the LW pig lineage in LU pigs have occurred. 

The results of ADMIXTURE are shown in Figure 3. K is an adjustable parameter representing the number of possible ancestral varieties. Through the calculation of the CV error, we obtained K = 2 as the best K value. In the case of K = 2, we could well-distinguish all three breeds. When the optimal K value was 2 for both of the platforms, we could distinguish between western and Chinese breeds. In the results of the SNP chip data, LU pigs, as a synthesized breed developed with YY and LW pigs, had a similar lineage to both of these breeds. This result verified the effectiveness of PCSSG in cultivating LU pigs, with a 50% YY pig lineage and a 50% LW pig lineage. However, in the GGRS results, LU pigs had a much more dominant LW pig lineage than YY pigs. This difference may have been caused by the high admixing of these 38 chosen LU pigs with LW pigs in the GGRS platform. 

### 3.3. Selection Signature Detection

The *F_st_* results of the SNP chip data are shown in Figure 4. The *F_st_* peak value of LW and LU pigs was less than 0.65, whereas the *F_st_* peak value of YY and LU pigs was up to 0.8, which showed that the relationship between LW and LU pigs was closer than that of YY and LU pigs. These results are consistent with those of ADMIXTURE. Because the LU pig is the derivation of LW and YY pigs, we can consider significant SNPs between LW and LU pigs as originating from YY pigs, and significant SNPs between YY and LU pigs as originating from LW pigs.

We ranked all SNPs in descending order by their *F_st_* values, and we selected the top 1% of SNPs for further genetic detection. Through gene annotation, we obtained 426 genes with significant differences between LU and LW pigs. Moreover, we obtained 782 significant genes between LU and YY pigs, including 3 major genes, namely LEPR (related to growth and fat accumulation), SCD (related to meat quality and fatty acid composition), and TBC1D1 (related to meat quality). These results are shown in Table 1. The three reported major genes are all related to body composition traits, and two of them (LEPR and SCD) are related to fat. These results indicated that LU pigs inherited certain genes related to meat quality from LW pigs. 

The *F_st_* results of the GGRS data are shown in Figure 5. The *F_st_* peak value of LW and LU pigs was less than 0.6, and the *F_st_* peak value of YY and LU pigs was as high as 1.0. In both of these platforms, the difference between LU and YY pigs was larger than that between LU and LW pigs. We used the same method described above to obtain 634 genes with differences between LU and LW pigs and 601 genes with differences between LU and YY pigs. We found three significantly different genes between LU and LW pigs, which were MYPN (related to body composition), KIT (related to coat color), and PPARD (related to ear size). We found three significantly different genes between LU and YY pigs, which were IGF2 (related to growth and fat accumulation), FTO (related to growth and fat accumulation), and KIT (related to hair color). These results are shown in Table 1. We used a total of 34 corresponding human orthologous genes for the GO, KEGG, and QTL annotation analysis. The results of the QTL enrichment analysis are shown in Figure 6.

### 3.4. Detection of Introgression

The rIBD distribution map of the SNP chip data is shown in Figure 7, in which a positive value represents a fragment that was introgressed from LW to LU pigs, and a negative value represents a fragment that was introgressed from YY to LU pigs. Overall, we found 26,677,332 bases and 27 genomic regions that were introgressed from LW to LU pigs, and we obtained 16,838,316 bases and 20 genomic regions that were introgressed from YY to LU pigs by linking significant consecutive rIBD fragments.

As shown in the figure, we obtained few sites from SNP chip, and the coverage of the whole genome was low. The results from the genetic mapping showed that 229 genes significantly introgressed from LW pigs, and 182 genes significantly introgressed from YY pigs. We found one major gene, SCD, to be related to meat quality and fatty acid profile.

The rIBD distribution map of GGRS data is shown in Figure 8. A total of 100,649,934 bases and 149 genomic regions introgressed from LW to LU pigs, and we obtained 38,666,133 bases and 94 genomic regions that introgressed from YY to LU pigs by linking significant consecutive rIBD fragments. Compared with the SNP chip data, GGRS provided higher coverage of the whole genome. The results of rIBD show that the rIBD fragments of LU pigs introgressed more from LW pigs than from YY pigs. Moreover, all significant rIBD fragments were from LW pigs. This result is completely different from the traditional understanding of the LU pig lineage. Combined with the results of PCA and ADMIXTURE, we inferred that the LU pigs that we chose for GGRS were likely deeply mixed with LW pigs.

## 4. Discussion

### 4.1. Population Structure

The results of the population admixture and PCA in both sequencing platforms showed that LU pigs were more scattered than the other two breeds. Thus, these results indicate that genetic differentiation between LW and LU pigs after PCSSG. In addition, the genetic distance between individuals of LU and LW pigs in the GGRS data was close, whereas the distance between individuals of LU and LW pigs in the SNP chip data was larger. Two possible reasons for this might be the difference between the sequencing platforms and population mixing caused by management confusion on the original Laiwu conservation farm.

### 4.2. Signatures Detected across Populations

The *F_st_* method has been widely used for exploring selection signatures between populations. The results of *F_st_* showed a significant difference between LU and YY pigs. Moreover, LU and LW pigs had fewer differences. Based on the results of gene mapping, we identified three major genes (MYPN, KIT, and PPARD.1) in the significantly different genes between LU and LW pigs. We mainly found three SNPs in the MYPN gene in western pig breeds, and these SNPs are significantly related to meat color, tenderness, and water-holding capacity in their body composition [23,24,25]. The KIT gene was confirmed to be related to coat color, and KIT has a high genetic diversity in the YY pig population, which also explains that the KIT gene has a high genetic diversity both in LU–LW and LU–YY pigs [26,27,28]. However, LW and LU pigs are both black with significant differences in KIT, which means that other genes play a key role in controlling their coat color. The PPARD gene is significantly related to ear size. LW pigs, as a local breed in China, have large and prone ears, and YY pigs have smaller and erect ears. This gene may play an important role in the regulation of the ear size of LW and YY pigs. Additionally, PPARD is associated with fat deposition, which may be the key influencing factor, resulting in lower intramuscular fat content in LU pigs compared with LW pigs [29,30].

Six major significantly different genes exist between LU and YY pigs, namely LEPR, SCD, TBC1D1, IGF2, FTO, and KIT. The LEPR gene is significantly correlated with backfat thickness and intramuscular fat [31,32]. The SCD gene is significantly related to fatty acid composition, which is one of the most important parameters in the evaluation of meat quality. The contents of IMF, saturated fatty acid, monounsaturated fatty acid, palmitoleic acid, and oleic acid in LW pigs are significantly higher than those in YY pigs [33]. The SCD gene is significantly related to palmitoleic acid (c16:1), stearic acid (c18:0), arachidic acid (c20:0), saturated fatty acid, and unsaturated fatty acid. The TBC1D1 gene is related to meat quality traits, and research has shown that this gene is associated with ham weight, backfat thickness, and lean meat percentage [34]. In 1999, the IGF2 gene was discovered to be bound to fat deposition, and was subsequently found to be significantly related to muscle development, lean meat rate, and other traits [35,36,37]. The FTO gene is related to fat content and obesity in humans, and the FTO gene in pigs is linked to muscle development and the rate of lean meat [38,39]. The KIT gene is significant not only between LU and LW pigs, but also between LU and YY pigs. This means that KIT may have complex mechanisms in the regulation of coat color, and the genetic diversity of KIT must be quite high.

The *F_st_* results of the two sequencing platforms showed that LU, as a composite breed of LW and YY pigs, has a high resemblance to LW but a lesser similarity to YY. We found that LU pigs not only inherited benefits from LW pigs, with high intramuscular fat and some meat quality traits, but they also inherited poor growth characteristics from LW pigs. In addition, LU pigs inherited fewer traits from YY, which were mostly traits relating to meat composition and appearance. We found that the PPARD gene is significantly different in LU and LW pigs. This can be one possible reason why LU pigs have larger ears and lower intramuscular fat content than LW pigs.

### 4.3. Detection of Introgression

Using the rIBD method in the SNP chip data, we obtained 27 significant segments that were introgressed from LW to LU pigs and 12 significant segments from that were introgressed from YY to LU pigs. We found the SCD gene in the selection signature as well as in LW pigs by gene mapping [40]. This gene is related to meat quality and fatty acid composition. In addition to from some fundamental biological functions, two pathways are linked to growth and reproduction from the deep section of the introgression of LW pigs, according to the results of the GO and KEGG annotation analyses. Regarding the GGRS data, we obtained 149 significant segments, including ESR1 (mainly related to reproduction), that were introgressed from LW to LU pigs [41]. However, we obtained few fragments and no significant segments that were introgressed from YY to LU pigs. By combining the results of PCA and ADMIXTURE, we inferred that these 38 LU pigs that we chose for GGRS from the original Laiwu conservation farm were likely deeply mixed with LW pigs during group management after cultivation on the original Laiwu conservation farm. In other words, under the improper management and mating system of the original Laiwu conservation farm, this population of LU pigs was far from its original state when it was successfully developed as a result of mixing too much LW pig blood in the process of breed conservation. In addition, errors may have occurred in the identification of these LU and LW pigs, some of which should be LU pigs but were wrongly marked as LW pigs and vice versa. From another point of view, this result may reflect the current situation of the conservation of LU pigs to some extent. The genetic resources of livestock and poultry in China are rich, but poor conservation management measures and the lack of awareness of breed protection have led to a mixture of different populations and a reduction in lineage purity, which seriously affects the sustainable development of livestock and poultry breeding. Based on the results of this experiment, improving the quality of the conservation and management of such farms is urgently required. 

The expected goal of breeding LU pigs using PCSSG is to improve growth performance and lean meat rate by introducing YY pig blood on the basis of maintaining the meat quality and reproductive characteristics of LW pigs. Regarding the results of this study, at present, LU pigs have achieved the expected breeding effect only in terms of reproductive characteristics. LU pigs have the advantages of LW pigs, such as their large litter size, which is consistent with the results of a previous study [42]. However, LU pigs have not achieved the expected breeding effect with respect to other traits. For example, we found that LU pigs inherited their intramuscular fat content and lean meat percentage from YY and LW pigs. Therefore, we inferred that the body composition of LU pigs is mediocre. The same circumstance applies to growth traits. In addition, the results of this study showed that LU pigs inherited the immune-related characteristics of YY pigs, and that LU pigs lost the strong stress resistance of LW pigs, in contrast to our expectations. From another point of view, we can also see the shortcomings and problems of PCSSG.

## 5. Conclusions

In this study, we analyzed the selection results of the LU pigs by the genetic structure of their population, their selection signature, and gene introgression analysis on the SNP chip and GGRS platform. These results of analysis verified our conjecture that the breeding effect of composite breeds of LU pigs has not fully meet the initial expectations from when it was developed, and these results further explain the problems and disadvantages of the traditional systematic crossbreeding method. We conclude that LU and LW pigs are genetically close together, and the LU pig population is highly admixed. Moreover, in terms of growth and fleshy bodies, the characteristics of the LW and YY pigs were simultaneously inherited, but differences remain between the two populations, which led us to the conclusion that the growth traits of the LU pigs are better than those of LW pigs but inferior to those of YY pigs, whereas the fleshy body composition traits were the opposite. In terms of reproduction, although LU pigs inherited the advantage of a high litter size from LW pigs, they did not inherit the good stress resistance of LW pigs. Therefore, breeding LU pigs using PCSSG combines some advantages of the two breeds, but its breeding effect is unsatisfactory compared with the breeding goal. This result also further reflects the limitations of PCSSG. In the future, we can use the marker information from whole-genome sequencing (WGS) to screen candidate parent populations and optimize corresponding parameters when determining the breeding scheme by using simulation analysis software. We can construct a selection index according to the kinship of specific genomes associated with different individual traits, and we can calculate the genome’s estimated breeding value (gEBV) and perform selection based on genome selection (GS). This may be a direction that we can follow to develop new genomic crossbreeding methods in the future. Moreover, the mixture of LW and LU pigs in the GGRS data also reflects the current conservation problems in China. However, LU pigs still have resource value. Although a large gap in growth performance remains between LU and YY pigs, to a certain extent, LU pigs incorporate the characteristics of the two parent breeds into the growth and quality of their bodies. By using a YY pig as a male parent and an LU pig as a female parent, we can conduct introductive crossing supplemented by genomic information to improve the breeding effect of LU pigs, thus improving their growth performance and maintaining their high meat quality and body composition characteristics.

## Figures and Tables

**Figure 1 genes-13-01969-f001:**
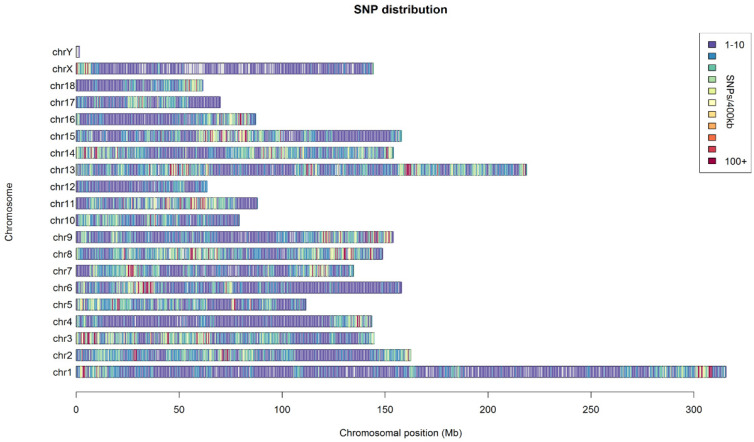
Distribution of the sequenced SNPs on all chromosomes. The y–axis represents chromosomes, and the x–axis represents the corresponding. Chromosomal position (Mb). Different colors of each 100 kb genome block denote the number of SNPs.

**Figure 2 genes-13-01969-f002:**
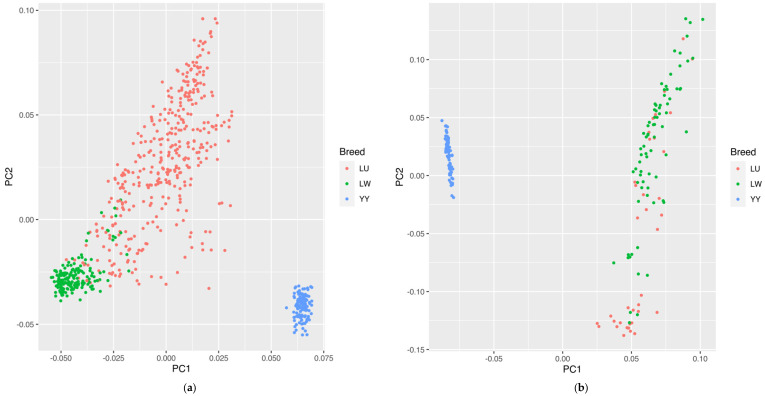
(**a**) PCA results of LU pig population by SNP chip. (**b**) PCA results of LU pig population by GGRS. Blue, red, and green circles represent YY, LU, and LW pigs, respectively.

**Figure 3 genes-13-01969-f003:**
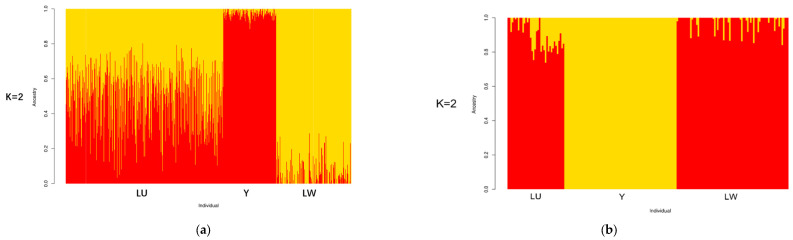
Structure results of ancestry compositions with the assumed number of ancestries at K = 2: (**a**) structure results of LU pig population by SNP chip; (**b**) structure results of LU pig population by GGRS.

**Figure 4 genes-13-01969-f004:**
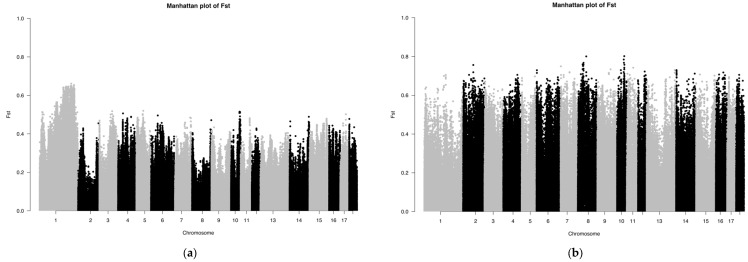
(**a**,**b**) Manhattan plot based on *F_st_* of LU–LW and LU–YY pigs by SNP chip.

**Figure 5 genes-13-01969-f005:**
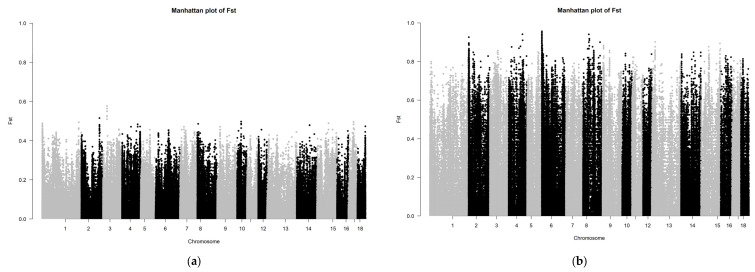
(**a**,**b**) Manhattan plot based on *F_st_* of LU–LW pigs and LU–YY pigs by GGRS.

**Figure 6 genes-13-01969-f006:**
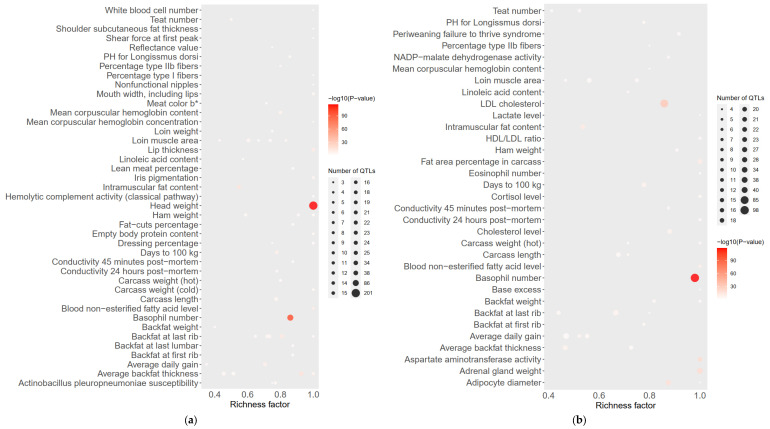
(**a**,**b**) Bubble plot based on *F_st_* of LU–LW pigs and LU–YY pigs by GGRS to exhibit QTL enrichment results. The significant phenotype of LU–LW is head weight and the significant phenotype of LU–YY is basophil number.

**Figure 7 genes-13-01969-f007:**
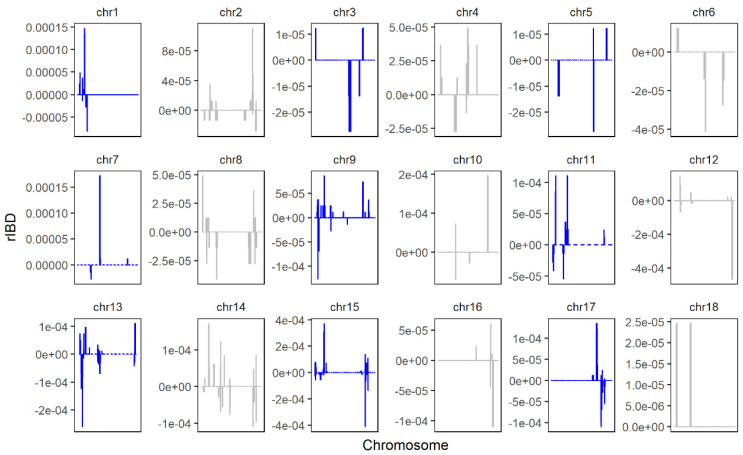
rIBD results of LU–LW and LU–YY pigs by SNP chip. Positive values represent rIBD between LU and LW pigs, which means the sequence was introgressed from LW to LU pigs. Negative values represent rIBD between LU and YY pigs, which means the sequence was introgressed from YY to LU pigs.

**Figure 8 genes-13-01969-f008:**
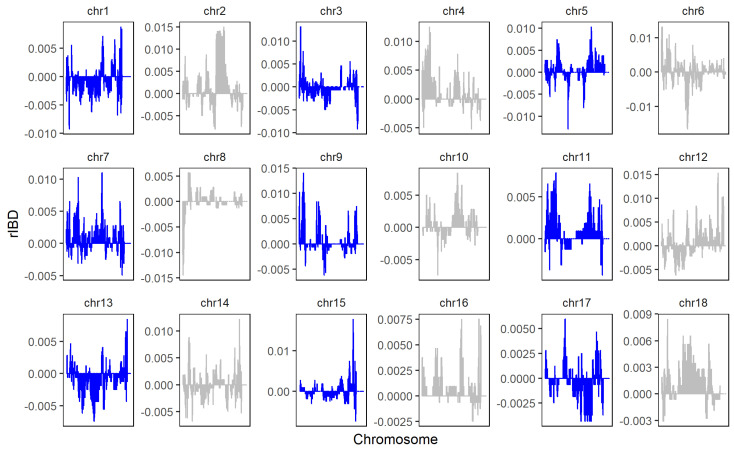
rIBD results of LU–LW and LU–YY pigs by GGRS. Positive values represent rIBD between LU and LW pigs, which means the sequence was introgressed from LW to LU pigs. Negative values represent rIBD between LU and YY pigs, which means the sequence was introgressed from YY to LU pigs.

**Table 1 genes-13-01969-t001:** Mapping results of significant genes of rIBD and *F_st_*.

*F_st_*	LU–LW	LU–YY
Number of Genes (Top 1%) (SNP chip)	426	782
Major Genes		LEPR (Growth, fatness), SCD (Meat quality, FA profile), TBC1D1 (Meat quality)
Number of Genes (Top 1%) (GGRS)	634	601
Major Genes	MYPN (Body composition), KIT (Coat color), PPARD (Ear size)	IGF2 (Growth, fatness), FTO (Growth, fatness), KIT (Coat color)
rIBD	LU–LW	LU–YY
Number of Genes (Top 1%) (SNP chip)	229	182
Major Genes	SCD (Meat quality, FA profile)	
Number of Genes (Top 1%) (GGRS)	229	182
Major Genes	ESR1 (Litter size)	

## Data Availability

The data presented in this study are available on request from the corresponding author. The data are not publicly available due to privacy restrictions.

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
