# Peer review of "Genomic Signatures Reveal Breeding Effects of Lulai Pigs"

_genes, 2022, doi:10.3390/genes13111969_

Round 1

Reviewer 1 Report

This an original paper focused on genomic signatures reveal breeding effect of Lulai pigs is interesting paper. The results of the obtained research were properly described and discussed. Despite the fact that the results may be slightly distorted by the use of different sequencing platforms, the authors with their findings appropriately point out the problems in breeding conservation farms and the problems in the protection of national breeds. I especially appreciate that in the conclusions the authors provide suggestions on how to improve the selection with regard to the breeding effect of Lulai pigs, which is very important from the farming economy.

Comments:

References are not given in the text according to the instructions for the author.

Author Response

Cover letter

October 19, 2022

The Editor

Genes Journal

Dear Ms. Zinnia Liu,

Thank you very much for giving us the opportunity to revise this manuscript. According to yours and the reviewers’ comments and suggestions, our manuscript has been revised carefully and completely. We have already corrected References in the introduction as well as added some details in chapters. These constructive comments and revision work improved this manuscript obviously.

We hope you will find that our contribution is suitable for publication in Genes Journal, and we are looking forward to your comments.

Sincerely yours,

Qishan Wang, Ph.D.

Department of animal breeding and genetics

Zhejiang University

E-mail: wangqishan@zju.edu.cn

Reviewer 2 Report

Lanes 66-67: ”Most of them are moderate(?) and lack of competitiveness, inheriting both the advantages and disadvantages of their parents.” Really? I didn’t know that! Are we talking about genetics?

Lanes 105-107: “In addition, we excluded reads that could not be mapped or were mapped to sex chromosomes as a result of their highly(?) inconsistency in heredity.” Sex chromosomes inconsistent in heredity? In 45 years of activity I never heard something like this.

Periods starting with “And”?

Pigs cultivated?

At least once in Methods the extended name of techniques and softwares could be used...

Do Authors know that, in the trials of terminal hybrid pigs, some or several crosses give, as expected according to the dominance effects, unsatisfactory results and are abandoned?

Are they sure to consider estimated breeding values as a good way of improvement of a cross? 

I have no doubts on the work and results obtained. However, should be presented in a correct way without opinions based on “personal genetics”, please.

Author Response

Cover letter

October 19, 2022

The Editor

Genes Journal

Dear Ms. Ziana Zhang,

Thank you very much for giving us the opportunity to revise this manuscript. According to yours and the reviewers’ comments and suggestions, our manuscript has been revised carefully and completely. These constructive comments and revision work improved this manuscript obviously.

The comments raised by reviewers have been responded point-by-point. In this revision, we have replaced and deleted the incoherent words in the chapters of abstract as well as introduction. In the English language and style, we revised the wrong use of English grammar. In results and discussion section, we removed some unreasonable quotations and descriptive utterances. Furthermore, we also corrected References in the bibliography.

We hope you will find that our contribution is suitable for publication in Genes Journal, and we are looking forward to your comments.

Sincerely yours,

Qishan Wang, Ph.D.

Department of animal breeding and genetics

Zhejiang University

E-mail: wangqishan@zju.edu.cn

Response to Reviewer 2 Comments

Point 1: Lanes 66-67: ”Most of them are moderate(?) and lack of competitiveness, inheriting both the advantages and disadvantages of their parents.” Really? I didn’t know that! Are we talking about genetics?

Response 1: Thanks for your reminder. Deleted. Please see P3/L66-67.

Point 2: Lanes 105-107: “In addition, we excluded reads that could not be mapped or were mapped to sex chromosomes as a result of their highly(?) inconsistency in heredity.” Sex chromosomes inconsistent in heredity? In 45 years of activity I never heard something like this.

Response 2: Thanks for your reminder. Revised. Please see P3/L105-107.

Point 3: Periods starting with “And”?

Response 3: Thanks for your reminder and I'm very sorry for my poor English writing. Revised throughout the manuscript.

Point 4: Pigs cultivated?

Response 4: Thanks for your reminder. Revised. Please see P10/L363.

Point 5: At least once in Methods the extended name of techniques and softwares could be used...

Response 5: Thanks for your reminder. Revised throughout the manuscript.

Point 6: Do Authors know that, in the trials of terminal hybrid pigs, some or several crosses give, as expected according to the dominance effects, unsatisfactory results and are abandoned?

Response 6: Thanks for your question. In the breeding process of Chinese breeding farms, due to the problem of sample size and measurement method, most selections are made based on phenotypes rather than dominance effects or additive effects. Although phenotypes are also affected by genotypes, how to evaluate the results of this selection method is what this article wants to discuss.

Piont 7: Are they sure to consider estimated breeding values as a good way of improvement of a cross?

Response 7: Thanks for your question.We are not examining the estimated breeding values.Because the estimated breeding value is not accurately evaluated due to the sample size and measurement problems during PCSSG, so we need to judge the results of PCSSG

Point 8: I have no doubts on the work and results obtained. However, should be presented in a correct way without opinions based on “personal genetics”, please.

Response 8: Thanks for your suggestion. Revision has been done throughout the manuscript so as to present results in a better way without the influence of personal opinions.

Round 2

Reviewer 2 Report

1) There are still problems for the english form;

2) I don't know whether the Abstract is too long; 

2) I am sorry, but I strongly disagree with several parts of conclusions. Problems of crossbreeding are known since the beginning of the use of crossbreeding and the following considerations appear to me as sold out.

Author Response

Cover letter

October 25, 2022

The Editor

Genes Journal

Dear Ms. Ziana Zhang,

Thank you very much for giving us the opportunity to revise this manuscript. According to yours and the reviewers’ comments and suggestions, our manuscript has been revised carefully and completely. These constructive comments and revision work improved this manuscript obviously.

The comments raised by reviewers have been responded point-by-point. In this revision, our manuscript has been edited by a profession English editing service (MDPI English Editing) suggested by Genes. In abstract section, we have replaced and deleted the incoherent words.

We hope you will find that our contribution is suitable for publication in Genes Journal, and we are looking forward to your comments.

Sincerely yours,

Qishan Wang, Ph.D.

Department of animal breeding and genetics

Zhejiang University

E-mail: wangqishan@zju.edu.cn

Response to Reviewer 2 Comments

Point 1: There are still problems for the english form;

Response 1: Thanks for your reminder. Edited by a profession English editing service (MDPI English Editing) suggested by Genes.

Point 2: I don't know whether the Abstract is too long;

Response 2: Thanks for your reminder. We replaced and deleted some incoherent words. Please see P2.

Point 3: I am sorry, but I strongly disagree with several parts of conclusions. Problems of crossbreeding are known since the beginning of the use of crossbreeding and the following considerations appear to me as sold out.

Response 3: Thanks for your reminder. We hope to propose better new methods by discussing the old method PCSSG and we are in progress.